# Biomechanical Considerations in the Orthodontic Treatment of a Patient with Stabilised Stage IV Grade C Generalised Periodontitis: A Case Report

**DOI:** 10.3390/bioengineering11040403

**Published:** 2024-04-19

**Authors:** Fung Hou Kumoi Mineaki Howard Sum, Zhiyi Shan, Yat Him Dave Chan, Ryan Julian Dick Hei Chu, George Pelekos, Tsang Tsang She

**Affiliations:** Faculty of Dentistry, The University of Hong Kong, 34 Hospital Road, Sai Ying Pun, Hong Kong SAR (000), Chinashanzhiy@hku.hk (Z.S.); george74@hku.hk (G.P.)

**Keywords:** periodontitis, pathological tooth migration, orthodontics, biomechanics

## Abstract

Orthodontic treatment of periodontally compromised patients presents unique challenges, including controlling periodontal inflammation, applying appropriate force, designing an effective dental anchorage, and maintaining treatment results. Deteriorated periodontal support leads to alterations in the biological responses of teeth to mechanical forces, and thus orthodontists must take greater care when treating patients with periodontal conditions than when treating those with a good periodontal status. In this article, we report the case of a 59-year-old woman with stabilised Stage IV grade C generalised periodontitis characterised by pathological tooth migration (PTM). The assessment, planning, and treatment of this patient with orthodontic fixed appliances is described. Moreover, the anchorage planning and biomechanical considerations are detailed. Specific orthodontic appliances were employed in this case to produce force systems for achieving precise tooth movement, which included a cantilever, mini-screws, and a box loop. Careful application of those appliances resulted in satisfactory aesthetic and functional orthodontic outcomes in the patient. This case highlights the importance of multidisciplinary collaboration in the treatment of patients with severe periodontitis and the potential for tailored biomechanical approaches in orthodontic treatment to furnish good outcomes.

## 1. Introduction

Periodontal disease is a common concern among adults worldwide and can lead to problems in both oral function and dentofacial aesthetics. The increasing number of adults seeking orthodontic treatment has led to a growing need for orthodontists to address periodontal problems alongside their primary focus on correcting malocclusions and improving aesthetics [1,2,3,4]. Moreover, the prevalence of pathological tooth migration (PTM) among patients with severe periodontal disease is high, with 30.03% to 55.80% experiencing this condition [5]. PTM is characterised by tooth displacement caused by the disruption of various factors (such as periodontal support) that maintain a tooth in its physiological position. PTM-related displacements may present as tilted or buccally flared posterior teeth, labially proclined anterior teeth with spaces, extruded incisors, or rotated teeth [6,7,8].

The process of orthodontic tooth movement involves a series of self-limiting inflammatory reactions, such as cellular, vascular, neural, and immunological responses. These reactions work together in a coordinated manner to stimulate bone resorption on the side under pressure and bone formation on the side under tension, ultimately leading to the desired movement of the tooth [9].

While periodontal diseases are triggered by plaque bacteria, the host’s response is thought to be a significant factor in the deterioration of bone and connective tissue. The presence of microbial antigens in plaque leads to inflammatory and immune reactions. The host’s response varies among individuals based on cytokine levels and the inflammatory cell response. IL-1β and PGE2 are key players in the process of soft tissue and bone resorption [10,11]. Additionally, mechanical force can elevate cytokine levels, such as IL-1β and PGE2, in the gingival crevicular fluid of teeth undergoing orthodontic movement [12,13]. The application of orthodontic forces to teeth creates areas of compression and tension in the periodontal ligament, affecting blood flow and cytokine release during tooth movement [14].

It is important to consider the potential impact of orthodontic tooth movement on patients with uncontrolled periodontal disease. Research suggests that self-limiting inflammatory reactions in these patients can progress to more severe inflammation, leading to breakdown of connective tissue and bone. Additionally, studies have shown that orthodontic tooth movement may contribute to destruction of connective tissue attachment at teeth with inflamed, infrabony pockets [15]. It is crucial to closely monitor and manage periodontal health in these cases to minimize potential risks and complications. 

It can be challenging to orthodontically treat patients with periodontal disease. The first challenge lies in controlling periodontal disease before beginning orthodontic treatment and maintaining regular periodontal care throughout treatment [6,7,8,16,17,18,19,20,21,22,23,24,25]. The second challenge involves obtaining adequate dental anchorage for orthodontic movement, as patients often present with multiple missing teeth and reduced periodontal support due to periodontal disease [26]. In such cases, the reinforcement of anchorage through alternative means, such as headgear, Nance buttons, bite blocks, and temporary anchorage devices (TADs), should be considered. Additionally, biomechanical factors, such as reductions in the magnitude of forces due to bone loss, and the need to adjust for an apically shifted centre of resistance (Cres) that affects the moment-to-force ratio (M/F) during teeth movement planning, must be taken into account [24,27]. The final challenge is related to compromised orthodontic treatment outcomes and the difficulty in maintaining good treatment outcomes in periodontal patients. For example, patients often present with multiple ‘black triangles’ and gingival recessions after orthodontic treatment, and these problems need to be solved via additional procedures such as interproximal reductions, fillings, and/or soft tissue grafts [28,29,30,31,32,33]. Upon completion of orthodontic treatment, teeth with less periodontal support are usually more mobile and thus must be retained using rigid materials. Consequently, retainer reviews should be conducted more frequently in these patients and possibly lifelong to ensure proper maintenance [3].

Successful orthodontic treatment in patients with periodontal disease requires a multidisciplinary approach involving collaboration among periodontists, orthodontists, and sometimes prosthodontists and maxillofacial surgeons. The following case report exemplifies the meticulous orthodontic assessment, planning, and treatment process for a patient with stabilised severe periodontal disease and PTM, with an emphasis on the intricacies of anchorage planning and biomechanical considerations of the fixed appliance designs. It sheds light on the critical factors that contribute to favourable treatment outcomes.

## 2. Materials and methods

### Case Report

A Chinese woman (Ms C) aged 59 years, 9 months with stabilised Stage IV grade C generalised periodontitis, formerly known as generalised aggressive periodontitis, was referred to the Orthodontic Department of the University of Hong Kong. The patient complained about upper and lower displaced teeth affecting her appearance and chewing function and stated that she wanted the alignment of her dentition to be improved, followed by prosthetic replacement of her missing teeth. She had unremarkable medical history.

Extra-orally, the patient presented with a convex profile, an acute nasolabial angle, and protrusive upper and lower lips in the sagittal dimension. In addition, the patient had a relatively normal lower facial height and a long upper-lip length in the vertical dimension. There was no obvious facial asymmetry in the transverse dimension (Figure 1).

Intra-orally, the molar relationship could not be evaluated. The canines were in Class I relationships. Tooth 21 was in Class I relationships, and teeth 12 and 22 were in Class III (anterior crossbite) relationships. The upper and lower incisors were proclined. The lower anterior teeth were canted down in a clockwise direction due to the attrition of teeth 31, 32, and 33. PTM-related tooth displacements included mesially tipped teeth 27, 37, and 47, over-eruption of tooth 16, and buccal displacement of teeth 24 and 25, which exhibited a scissor-bite tendency to the opposing teeth. There were generalised recessions and mobilities. Multiple teeth were missing, namely teeth 17, 14, 11, 26, 36, and 46. There was recurrent caries along the margin of the ceramo-metal crown (CMC) on tooth 21. In addition, there was enamel hypoplasia on the upper and lower anterior teeth. Moreover, there was clinical attachment loss up to 8 mm on teeth 16 and 18, accompanied by generalised horizontal bone loss (Figure 1). 

This patient underwent non-surgical periodontal treatment, including root surface debridement and oral hygiene instructions. In order to let the patient have a more comfortable non-surgical therapy, non-surgical periodontal treatment was performed by quadrant weekly. With this quadrant-by-quadrant approach, the periodontal colleague would have more chairside opportunities to reinforce the patient’s oral hygiene, especially a patient presenting with severe malalignment, which further complicate the oral hygiene performance. After periodontal re-evaluation, persistent deep pockets remained on teeth 18, 16, and 21, respectively. Tooth 18 was to be extracted before orthodontic treatment, while teeth 16 and 21 had undergone open flap debridement and their pocket depth were further reduced. No antibiotics were used to control her periodontal disease prior to the orthodontic treatment. The baseline periodontal charting prior to the periodontal treatment taken on 28 August 2019 and the periodontal charting taken on the supportive periodontal care appointment on 18 April 2020 are shown in Figure 2. With the challenging situation in her oral cavity, after her periodontal status had been stabilised, she agreed to go for orthodontic treatment, to improve the aesthetics, occlusion for function, and alignment for later periodontal maintenance. During the orthodontic treatment, the patient was strictly reviewed and maintained by a periodontist colleague for a 3-month interval.

The pre-treatment periodontal charting taken on 28 August 2019 showed that the patient had interdental clinical attachment loss up to (1) >8 mm on teeth 18, 17, 16, 12, 11, 21, 28, 43, 44, 45, and 47; (2) 5–7 mm on teeth 15, 13, 22, 23, 24, 25, 27, 38, 37, 31, 41, and 48 (Figure 2). The patient received non-surgical periodontal therapy, including root surface debridement and oral hygiene instructions. After periodontal reevaluation, persistent deep pockets remained on teeth 18, 16, and 21, respectively. Teeth 16 and 21 had undergone open flap debridement and their pocket depth was further reduced. Before commencement of the orthodontic treatment, only teeth 18 and 16 had clinical attachment loss of >8 mm. Both teeth were planned to be extracted in the future (Figure 2).

A panoramic radiograph showed generalised horizontal bone loss, multiple missing teeth, and tilted premolars and molars. There was also an asymptomatic supernumerary tooth apical to teeth 12 and 13 (Figure 3).

Cephalometric tracing revealed a Class I skeletal relationship with a prognathic maxilla and mandible. The upper and lower incisors were proclined, and the interincisal angulation was reduced. In addition, the lower incisors were anterior to the A-point–pogonion (Apo) line (Table 1 and Figure 4).

In summary, the patient was a 59-year- and 9-month-old Chinese woman with stabilised Stage IV grade C generalised periodontitis characterised by (1) generalised clinical attachment loss of more than or equal to 5 mm. (2) Teeth 16 and 18, which had clinical attachment loss of 8 mm, were assessed to have a poor prognosis. (3) Multiple teeth, including teeth 17, 14, 11, 26, 36, and 46, were lost due to periodontitis, and (4) there were PTM-related tooth displacements, including mesially tipped teeth 27, 37, and 47, over-eruption of tooth 16, and buccal displacement of teeth 24 and 25. Ms. C’s upper and lower incisors were proclined on a bimaxillary protrusive Class I skeletal base. She had a convex profile, and a protrusive lower lip. The canine relationships were Class I. Teeth 12 and 22 were in crossbite, and teeth 24 and 25 exhibited a scissor-bite tendency. There was a supernumerary tooth apical to teeth 12 and 13. The lower incisors were positioned anterior to the Apo line. There was caries at the CMC margin of tooth 21. The upper and lower anterior teeth were hypoplastic. The lower anterior teeth were canted down on the left side due to enamel wear on the incisal edges of teeth 33, 32, and 31.

## 3. Treatment Objectives

The treatment objectives were to (1) accept the basal relationship; (2) retract the upper central incisors’ position by 2 mm to the anterior–posterior (AP) position at teeth 12 and 22; (3) correct the scissor-bite involving teeth 24 and 25; (4) level the curve of Spee; (5) align the teeth and harmonise the arches; (6) redistribute the spaces for future replacement of missing teeth; and (7) improve the lip profile.

## 4. Treatment Progress

### 4.1. Pre-Orthodontic Dental Treatment Preparation

Before beginning orthodontic treatment, periodontal disease must be controlled. The patient received non-surgical periodontal therapy, including root surface debridement and oral hygiene instructions. After periodontal reevaluation, persistent deep pockets remained on teeth 18, 16, and 21, respectively. Tooth 18 had a poor prognosis and uncontrolled inflammation and thus was extracted. Teeth 28, 38, and 48 were also extracted, as they hindered the distal crown tipping of teeth 27, 37, and 47. Teeth 16 and 21 had undergone open flap debridement and their pocket depth was further reduced. Tooth 16 also had a poor prognosis but was not extracted, as its inflammation was controlled, and it was to be used as an anchorage during orthodontic treatment. Accordingly, tooth 21 was treated via root canal, and the CMC was replaced with a composite temporary crown. Finally, the asymptomatic supernumerary tooth was kept under review throughout the orthodontic treatment.

### 4.2. Anchorage Design

Anchorage planning was divided into the transverse dimension, the vertical dimension, and the sagittal dimension. In the transverse dimension, tooth 16 was to be used as free anchorage to rotate the crowns of teeth 24 and 25 crown palatally. Free anchorage is anchorage for which no ‘price’ has to be paid in terms of undesirable force on teeth belonging to the anchorage unit. In principle, reactive forces are transferred to teeth that are to be extracted according to the treatment plan, such that there are no adverse effects on the teeth that are to remain in the arches following treatment [35]. Tooth 16 was to be extracted after teeth 24 and 25 had been uprighted palatally. In the vertical dimension, teeth 24 and 25 were to be intruded, and tooth 16 was to be extruded with the force system used to upright teeth 24 and 25. In the sagittal dimension, the plan was to retract the upper and lower arches by 2 mm with TADs. The planned post-treatment AP dental position of the upper and lower incisors is illustrated in the visual treatment objectives (VTOs) (Figure 5). Occlusograms were used to illustrate the two final arches passing through the planned post-treatment dental contact points (Figure 6) [36]. The force system is detailed in the following section on biomechanical planning.

### 4.3. Biomechanical Planning

Regarding the biomechanical approach for uprighting teeth 24 and 25, a one-couple force system, also known as cantilever mechanics, was used, as it enabled the generation of a statically determinate force system to achieve the desired tooth movement [37]. The appliance used had two connections. On the left side, there was a palatal sheath (0.036″ × 0.072″) soldered onto a cobalt–chromium (CoCr) plate bonded to the palatal side of teeth 24 and 25. A 0.036″ beta-titanium (β-Ti) wire was folded and inserted into the palatal sheath, creating a two-point contact that generated palatal crown torque and intrusive force. On the right side, there was a palatal cleat welded to the molar band of tooth 16. A 0.036″ β-Ti wire was formed with a closed loop, which was tied to the palatal cleat with a steel ligature wire to create a one-point contact that generated an extrusive force. In Figure 7, this activation force system is illustrated by the blue arrows, while the deactivation force system is illustrated by the red arrows. The photographs before and after cantilever mechanics were used to upright the premolars are shown in Figure 8.

Regarding the biomechanical approach employed for retracting the upper and lower incisors, buccal shelf mini-screws were used and were subjected via elastics to a retractive force from teeth 13, 23, 33, and 43 (Figure 9).

During the space closure stage, the power chain on tooth 35 dislodged and the tooth relapsed. Thus, we adopted a box loop; we applied a 0.018” stainless-steel wire to tooth 35 to simultaneously tip its crown mesially and rotate its crown mesio-bucally. According to Burstone, [38] teeth 34 and 35 exhibited Class III geometry in both the occlusal and the buccal view (Figure 10). In the occlusal view, with the box loop, there were moments that rotated teeth 34 and 35 mesio-buccally. In contrast, the powerchain on teeth 34–44 created a mesio-lingual moment on tooth 34 that cancelled out the mesio-buccal moment created by the box loop. In the buccal view, there were moments that tipped the crowns of teeth 34 and 35 mesially. In the occluso-gingival plane, the wire end was deflected gingivally before insertion into the bracket of tooth 35. In the buccal–lingual plane, the end of the wire was deflected buccally before insertion into the bracket of tooth 35 (Figure 10).

### 4.4. Additional Dental Treatments

Additional dental treatments after orthodontics treatment included regular supportive periodontal care supplied by a periodontist colleague. Moreover, bone augmentation and implant replacement of missing teeth 17, 16, 14, 26, 36, and 46 were performed by an oral and maxillofacial surgeon and a prosthodontist colleague. A three-unit full ceramic bridge spanning teeth 12–21 was constructed to replace missing tooth 11, and composite veneers were constructed for all the hypoplastic teeth by the prosthodontist colleague. The orthodontist colleague was responsible for regular review of the retainers and the supernumerary tooth.

### 4.5. Retention

Post-treatment photographs are shown in Figure 11, and near-end-of-treatment radiographs and superimpositions are shown in Figure 12, Figure 13 and Figure 14. The canine and incisal relationships were in Class I. There was a fixed labial retainer spanning teeth 12–21. There were fixed palatal retainers spanning teeth 12–13 and teeth 22–23. There was also a fixed lingual retainer spanning teeth 33–43 together with upper and lower temporary vacuum-formed retainers. The fixed labial retainer originating from teeth 12–21 was later replaced with a temporary composite bridge, and the fixed palatal retainer originating from teeth 12–13 was replaced with a composite splint (Figure 15). Composite veneers were installed on all the premolars, canines, and incisors. New temporary upper and lower vacuum-formed retainers were given to the patient. In future, we plan to place a fixed palatal retainer on teeth 13–23 and install upper and lower Hawley retainers after prosthetic replacement of all missing teeth. Figure 16 presents the patient’s pre-treatment and post-treatment occlusal contacts. The occlusion was unevenly distributed before orthodontic treatment with multiple teeth in heavy occlusion. In contrast, it was more evenly distributed after orthodontic treatment, with fewer teeth in heavy occlusion.

## 5. Discussion

This case report provides valuable insights into the assessment, planning, and treatment of a patient with stabilised Stage IV grade C generalised periodontitis accompanied by pathological tooth migration (PTM), highlighting the special biomechanical designs adopted. Despite meticulous planning, several challenges were encountered during treatment.

**(1)** 
**Challenges and precautions in maintaining stable periodontal health in patient with Stage IV grade C generalised periodontitis throughout the orthodontic treatment**


It can be challenging to orthodontically treat patients with Stage IV grade C generalised periodontitis. The main challenge from the periodontal perspective was to control the periodontal disease before beginning orthodontic treatment and to maintain regular periodontal care throughout treatment. For this patient presenting with Stage IV grade C generalised periodontitis, her oral hygiene status was evaluated before, during, and following orthodontic treatment.

(1.1)Before beginning the orthodontic treatment, clinical and radiographic periodontal assessment was performed by the periodontal colleague to confirm that the patient’s periodontal condition had been stabilised after the series of non-surgical and surgical periodontal therapies. Clinical examination included full mouth probing, tooth mobility assessment, and measurement of the amount of gingival recession. The full mouth plaque and bleeding on probing scores were below 25% before beginning the orthodontic treatment [39,40].(1.2)During orthodontic treatment, oral hygiene instruction and motivation was performed after placement of the orthodontic appliance. The first stage of orthodontic treatment was to upright the buccally flared teeth 24 and 25 using the cantilever mechanics. In this stage, we placed a CoCr plate covering the palatal surface of teeth 24 and 25. The bulkiness of the CoCr plate will hinder the patient’s cleaning efficiency and predispose the area to food entrapment. Emphasising the importance of meticulous oral hygiene in this area became imperative. During every visit, the oral hygiene status of the patient was checked by the orthodontic colleague. Regular periodontal recall appointments were also carried out every three months by the periodontal colleague [39,40]. In the second stage of the orthodontic treatment, the upper and lower teeth were bonded with fixed labial orthodontic brackets. The usage of fixed labial orthodontic brackets and wire created a difficult situation for the patient in performing optimal oral hygiene. The gingival margins and interdental areas were often blocked by the orthodontic appliances. Additional oral hygiene aids including waterfloss and superfloss were used when the patient could not clean with regular aids, like a toothbrush, ID brush, or single tuft brush. At some particular point in time, gingivae may present with more marginal inflammation, thus slightly increased pocket depth due to edema. As precautions, supportive periodontal care was carried out strictly every 3 months, and all the sites with orthodontic appliances were deplaqued thoroughly, to maintain a reasonable periodontal health to prevent periodontal flare up during orthodontic treatment.(1.3)Following orthodontic treatment, clinical periodontal evaluation combined with radiographic examination once a year will be performed [39,40].

**(2)** 
**Challenges and precautions in treating orthodontic case with supernumerary tooth**
(2.1)The patient refused to surgically extract the supernumerary tooth as the supernumerary tooth was not associated with any pathology in the patient’s upper jaw for more than 59 years. The decision to keep the supernumerary tooth did affect the planned retraction of upper incisors. This meant that we could not retract the upper incisors as much as we had planned to based on the VTO. As a precaution, radiographs were taken regularly during the orthodontic treatment to ensure that there was no pathology with the supernumerary tooth and the supernumerary tooth did not affect the movement of the upper anterior teeth. As the mid-treatment orthopantomogram (Figure 12) showed that the root apex of tooth 23 was very close to the supernumerary tooth, we decided to stop further retracting the upper teeth.

**(3)** 
**Challenges in planning the orthodontic biomechanics when treating patient with Stage IV grade C generalised periodontitis**


There were a few biomechanical challenges we faced when treating this patient with Stage IV grade C generalised periodontitis.

(3.1)The first biomechanical challenge involves obtaining adequate dental anchorage for orthodontic movement, as patients often present with multiple missing teeth and reduced periodontal support due to periodontal disease [26]. In this case, the reinforcement of anchorage through alternative means, such as temporary anchorage devices (TADs) and using tooth 16 as free anchorage, was performed.(3.2)The second biomechanical challenge involves controlling the magnitude of force, which should be reduced proportionally to the alveolar bone height [3,23,24].(3.3)The third biomechanical challenge relates to the apically shifted centre of resistance (Cres) of a periodontally compromised tooth, as the alveolar bone height is decreased, which increases the M/F [24,27]. Consequently, there is an increased tendency for such teeth to tip and rotate excessively in response to orthodontic forces, which are applied at the regular bracket level, leading to unwanted tooth movement and excessive friction during space closure with sliding mechanics.(3.4)During the treatment of this patient with Stage IV Grade C generalised periodontitis, we addressed the biomechanical challenges outlined in lists 3.1 to 3.3 at four specific time points.
(3.4.1)As teeth 24 and 25 were significantly buccally displaced, sectional cantilever mechanics were used rather than direct bonding of the entire upper arch. There were two reasons for this. The first reason was to avoid creating unfavourable force systems on the other teeth due to the buccally displaced positions of teeth 24 and 25. Bonding brackets onto all upper teeth and engaging them with a straight wire could have resulted in an abnormal or skewed arch form. The second reason was that tooth 16 had a poor prognosis and was to be extracted in the future. Therefore, we decided to use tooth 16 as free anchorage to bring teeth 24 and 25 into the planned archform with cantilever mechanics, before proceeding with straight wire mechanics.(3.4.2)During the alignment and levelling stage, tooth 35’s crown was displaced distally and rotated disto-bucally, so a box loop capable of delivering the desired magnitude of force and precisely regulating the force direction was used. The box loop increased the total amount of wire between brackets of teeth 34 and 35, which produced a reduced load–deflection ratio and a greater range of action than any other loop type [41]. Furthermore, as the box loop was composed of a series of vertical and horizontal levers contoured to provide a short section of archwire that was freely movable in all planes in which it was activated, we used it to tip the crown of tooth 35 mesially and rotate tooth 35 mesio-lingually [41]. Moreover, sufficient anchorage was provided by the relatively rigid, continuous wire portion adjacent to the box loop, spanning from teeth 34 to 45, which effectively differentiated the active and reactive units.(3.4.3)As discussed in list 3.3, periodontally compromised teeth would tip excessively in response to orthodontic force which is applied at the regular bracket level. Therefore, if the aim is to achieve bodily retraction of incisors, it is best to bond the brackets as cervically as possible. However, since the pre-treatment upper and lower incisors were proclined, we aimed to achieve ‘uncontrolled tipping’ during retraction, and thus, the brackets were bonded at the regular positions on the upper and lower incisors [42].(3.4.4)The attempted placement of mini-screws on the upper alveolar bone resulted in dislodgement due to the low sinus floor. This was caused by sinus peumatization which was a continuous physiological process that increased the volume of the paranasal sinuses [43]. To address this, lower buccal shelf mini-screws were used instead to retract the upper arch with Class I elastics.

## 6. Conclusions

Overall, successful management of periodontally compromised patients undergoing orthodontic treatment necessitates a holistic evaluation of periodontal and orthodontic factors, interdisciplinary collaboration in treatment planning, and meticulous attention to anchorage and biomechanics to ensure optimal treatment outcomes.

The overall orthodontic therapy summary of this patient is illustrated in Figure 17.

## Figures and Tables

**Figure 1 bioengineering-11-00403-f001:**
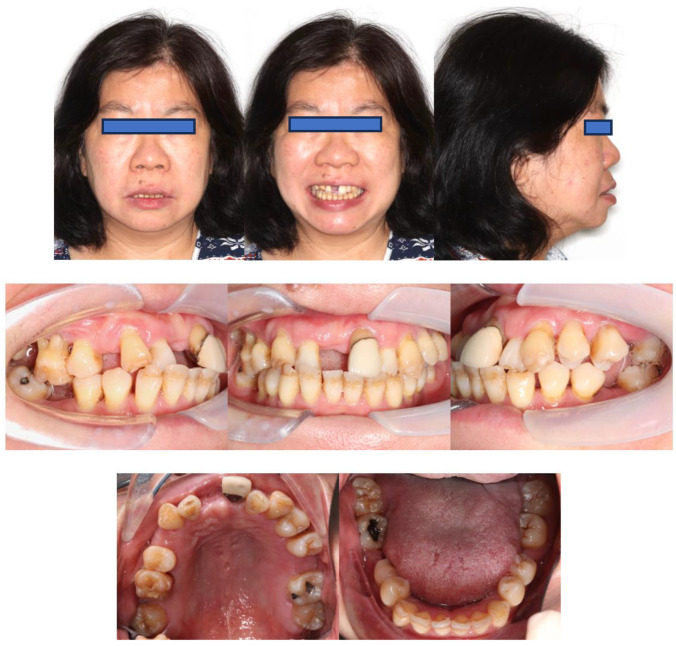
Pre-treatment photographs.

**Figure 2 bioengineering-11-00403-f002:**
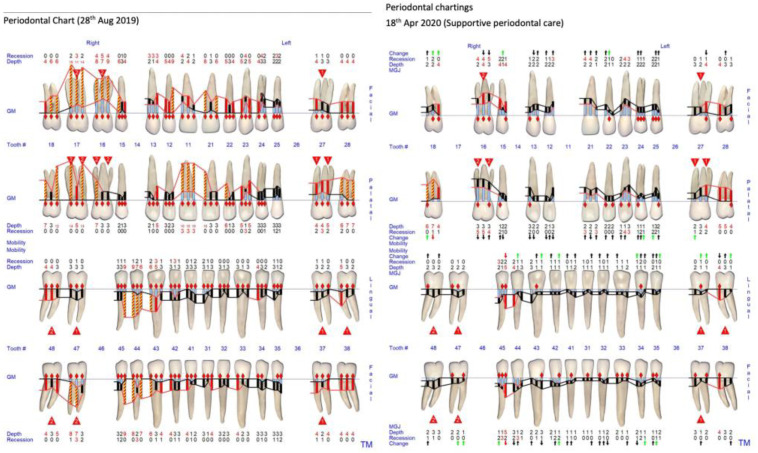
Baseline periodontal charting taken on 28 August 2019 and periodontal charting taken on 18 April 2020 at patient’s supportive periodontal care appointment before beginning of the orthodontic treatment. The black arrow pointing up means improvement of the CAL of less than 1 mm. The green arrow pointing up means improvement of the CAL of >1 mm. The black arrow pointing down means worsening of the CAL of less than 1 mm. The red arrow pointing down means worsening of the CAL of >1 mm.

**Figure 3 bioengineering-11-00403-f003:**
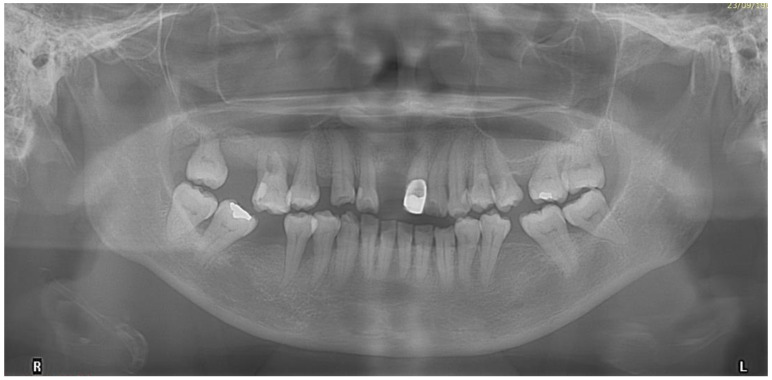
Pre-treatment panoramic radiograph.

**Figure 4 bioengineering-11-00403-f004:**
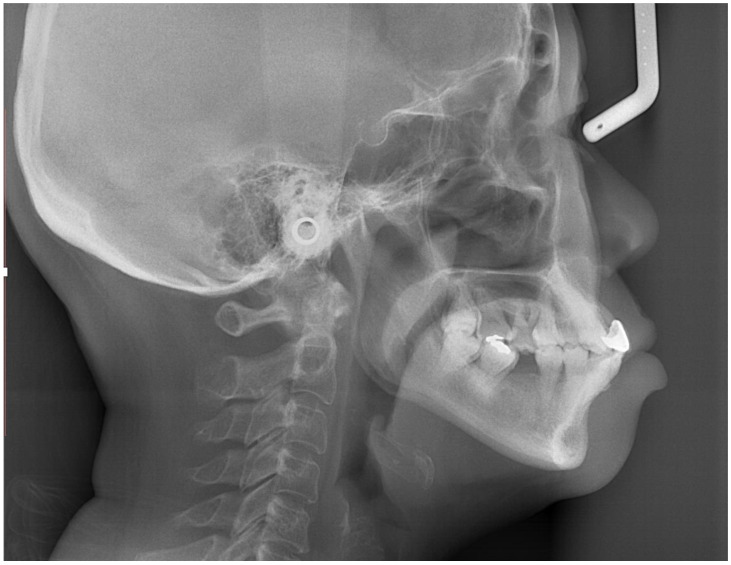
Pre-treatment lateral cephalometric radiograph.

**Figure 5 bioengineering-11-00403-f005:**
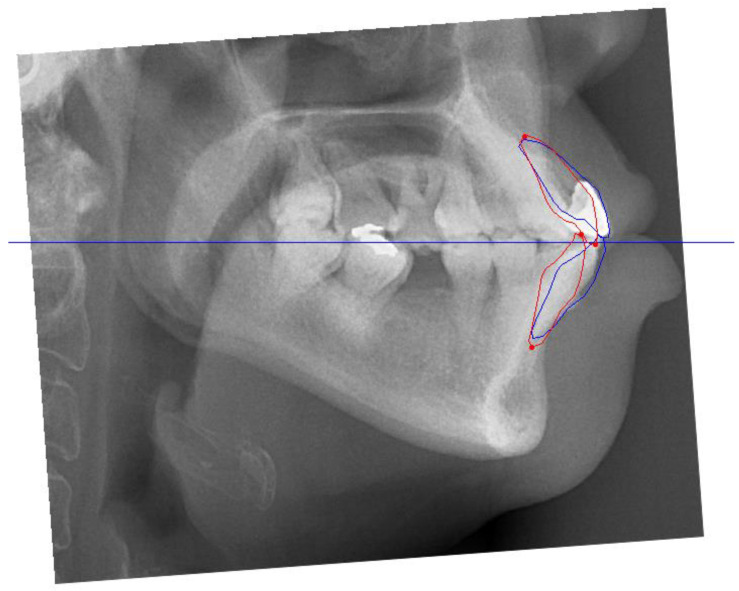
Visual treatment objectives (VTOs) showing the planned post-treatment anterior–posterior dental positions of the upper and lower incisors. The blue lines indicate the pretreatment positions of the incisors, and the red lines indicate the planned post-treatment positions. The blue horizontal line represents the occlusal plane of the patient.

**Figure 6 bioengineering-11-00403-f006:**
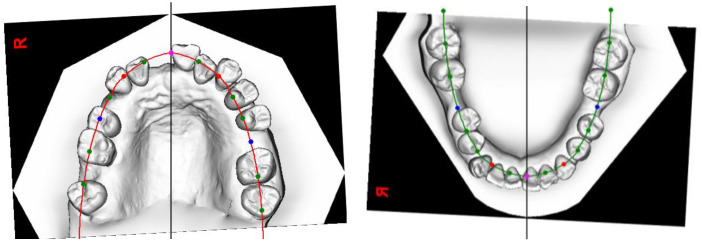
Occlusogram illustrating the patient’s upper and lower final arches passing through the planned dental contact points. The mesio-distal width of each tooth is aligned along the constructed arches. The blue dots indicate the planned post-treatment distal contact points of the second premolars. The green dots indicate the planned post-treatment mesial contact points of the second molars, premolars, and lateral incisors. The red dots indicate the planned post-treatment mesial contact points of the canines. The pink dots indicate the planned post-treatment contact points of the central incisors.

**Figure 7 bioengineering-11-00403-f007:**
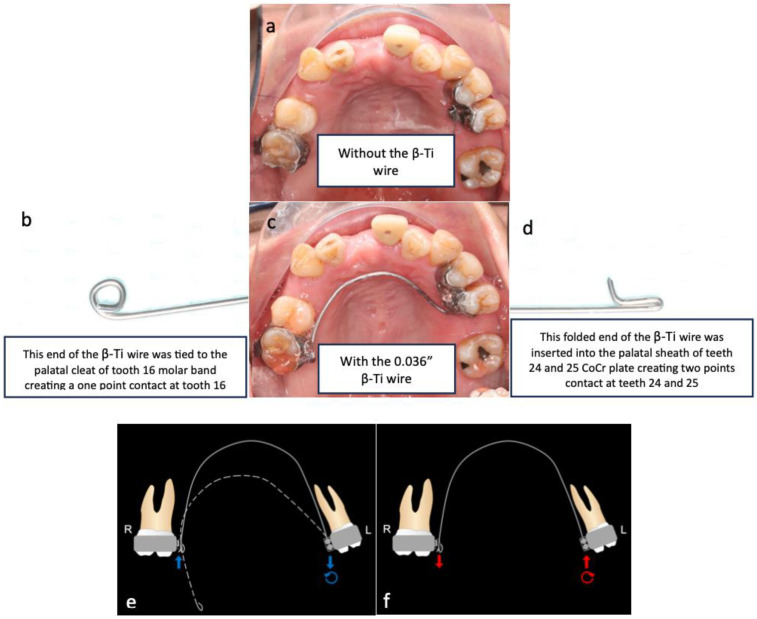
(**a**). Occlusal view of the cantilever without the β-Ti wire. (**b**–**d**). Occlusal view of the cantilever with the 0.036″ β-Ti wire tied onto the tooth 16 palatal cleat to create a one-point contact. The other end of the cantilever was a folded 0.036″ β-Ti wire that was inserted into the palatal sheath of the CoCr plate of teeth 24 and 25 to form a two-point contact. (**e**,**f**) demonstrate the activation force system (blue arrows) and deactivation force system (red arrows) of the cantilever, respectively. The deactivated shape of the cantilever is indicated by the dotted line, and the cantilever was activated by pushing the right side up (until it reached the solid line) to engage the β-Ti wire on the tooth 16 palatal cleat, as shown (**e**). During deactivation, the cantilever generated an extrusive force on tooth 16, an intrusive force on teeth 24 and 25, and a moment that rotated the teeth 24 and 25 crowns palatally (**f**).

**Figure 8 bioengineering-11-00403-f008:**
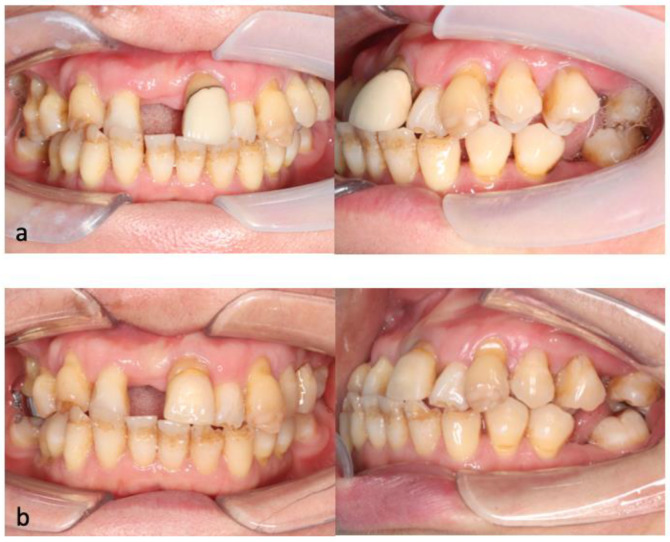
(**a**). Photographs of the patient before uprighting of teeth 24 and 25. (**b**). Photographs of the patient after uprighting of teeth 24 and 25.

**Figure 9 bioengineering-11-00403-f009:**
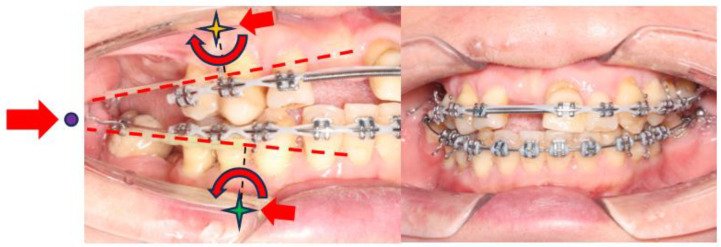
The maxillary premolar and premolars were tied as one segment using a powerchain, and the mandibular premolars and premolars were tied as one segment using another powerchain. The maxillary and mandibular arches were retracted by applying a retractive force from the upper and lower canines to a mini-screw inserted into the mandibular buccal shelf on both sides (purple circles). The upper arch was retracted using Class I elastics (3/16″, 3.5 oz) linking the upper canines to the corresponding lower mini-screws on each side. Similarly, the lower arch was retracted using powerchains from the lower canines to the mini-screws on the same sides. (The lines of action are indicated by red dotted lines.) The application of such mechanics meant that the deactivation force system at the Cres of the upper (yellow-starred) and lower (green-starred) arches consisted of distal forces (small red arrows) on both arches, with a clockwise moment on the upper arch and an anticlockwise moment on the lower arch (curved red arrows). The deactivation force system at the mini-screw consisted of a mesial force (large red arrow). As each mini-screw was fixed, only the upper and lower teeth were retracted and tipped distally.

**Figure 10 bioengineering-11-00403-f010:**
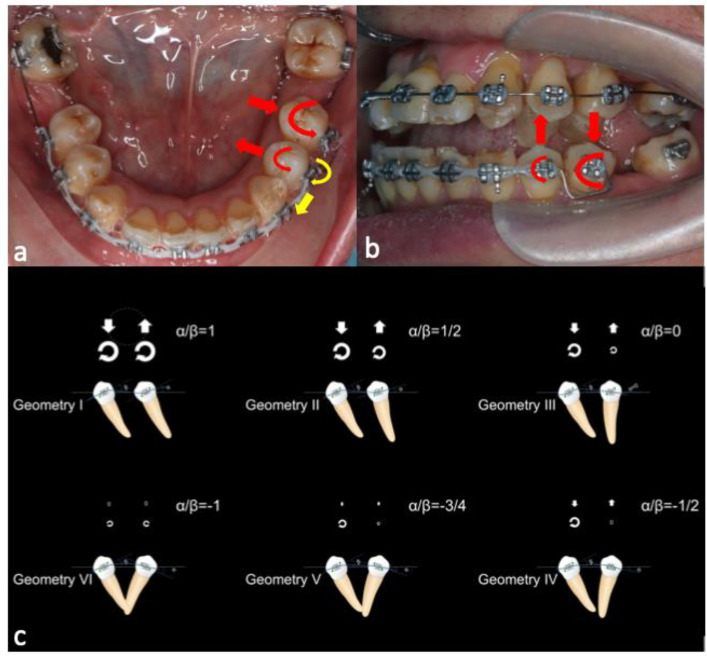
Teeth 34 and 35 were in Class III geometry in both the occlusal and the buccal view. (**a**,**b**) From the occlusal view, with the box loop, there were moments that rotated teeth 34 and 35 mesio-buccally (red arrows). In contrast, the powerchain on teeth 34–45 created a mesio-lingual moment on tooth 34 (yellow arrow) that cancelled out the mesio-buccal moment created by the box loop. (**a**) In the buccal view, there were moments that tipped the crowns of teeth 34 and 35 mesially (red arrow). (**b**) A schematic illustration of the six geometries is shown in (**c**).

**Figure 11 bioengineering-11-00403-f011:**
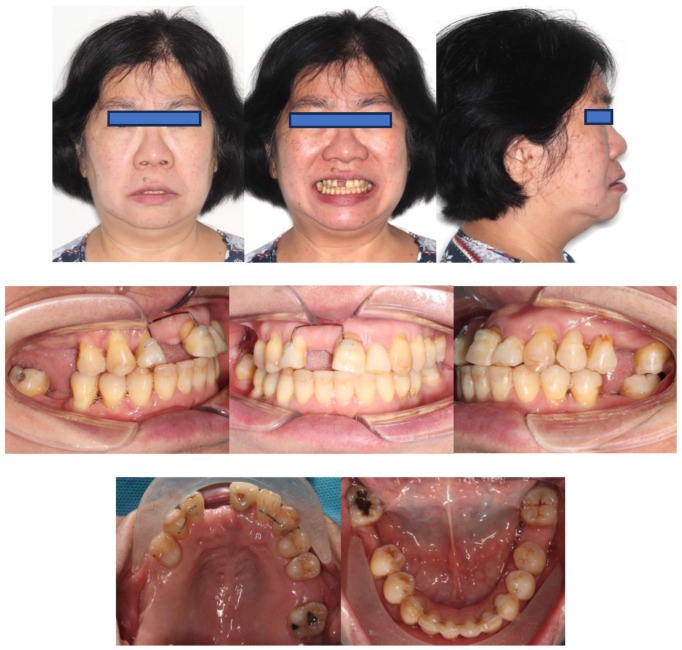
Post-orthodontic treatment photographs.

**Figure 12 bioengineering-11-00403-f012:**
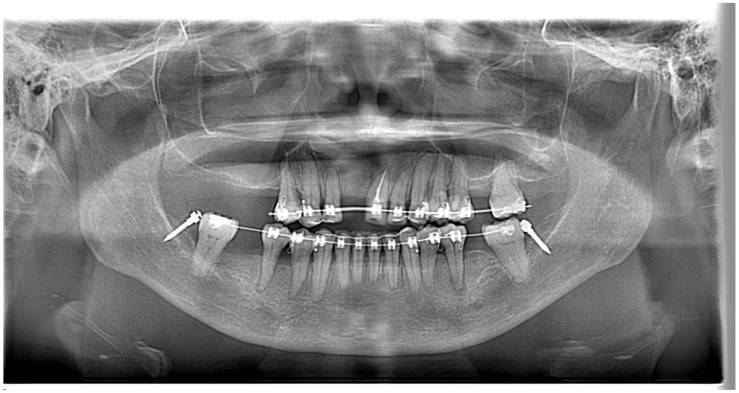
Near-end-of-treatment panoramic radiograph.

**Figure 13 bioengineering-11-00403-f013:**
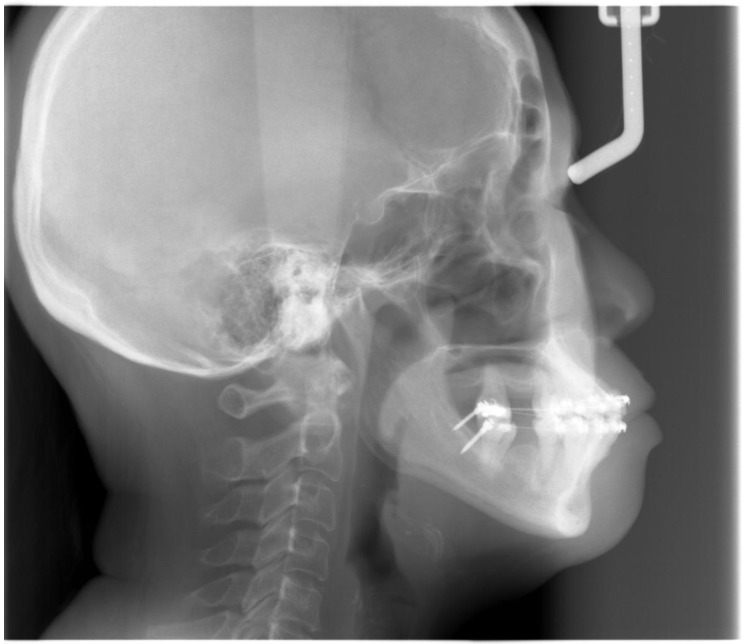
Near-end-of-treatment lateral cephalometric radiograph.

**Figure 14 bioengineering-11-00403-f014:**
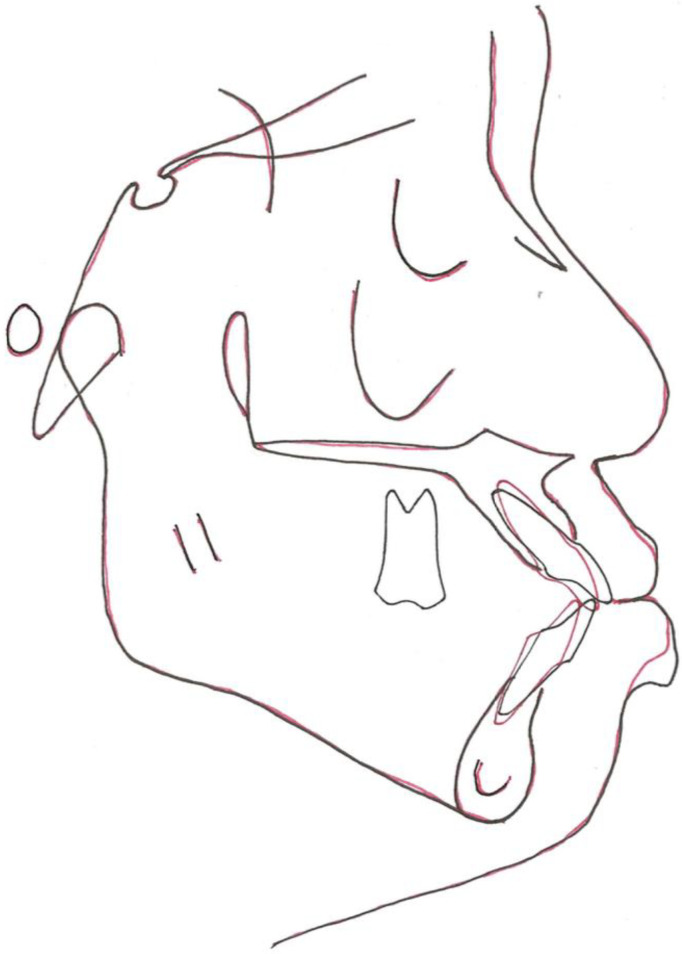
Superimposition of the pre-treatment and post-treatment lateral cephalometric radiographs.

**Figure 15 bioengineering-11-00403-f015:**
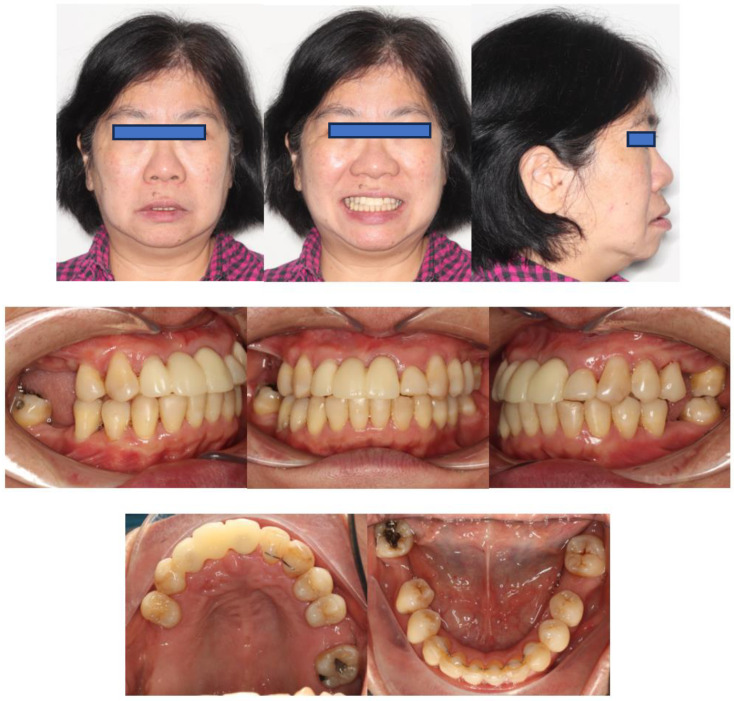
Follow-up review photographs after prosthetic replacement of tooth 11 with a temporary composite bridge from teeth 12–21 and with composite veneers on upper and lower premolars, canines, and incisors.

**Figure 16 bioengineering-11-00403-f016:**
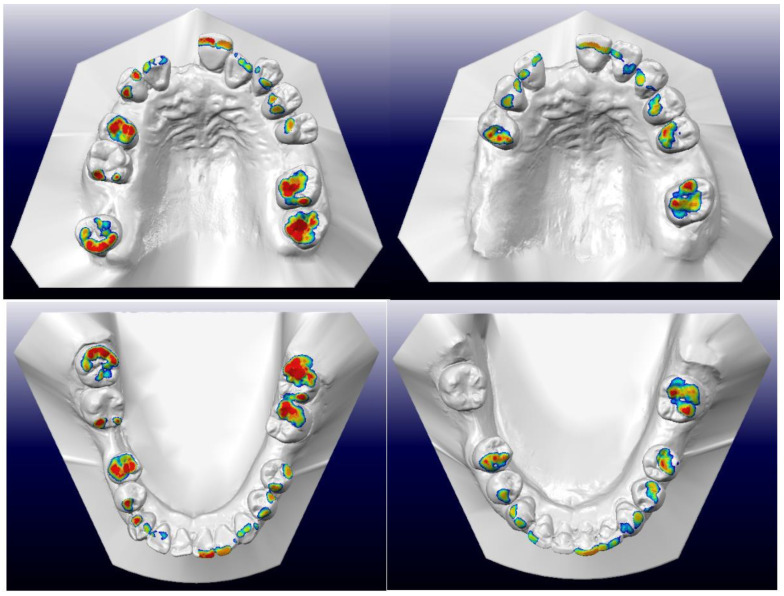
Left side: pre-treatment upper and lower occlusal contacts. Right side: post-treatment upper and lower occlusal contacts. Red indicates heavy occlusal contacts, whereas blue and green indicate light occlusal contacts. Orthodontic treatment increased the evenness of the distribution of the occlusal contacts.

**Figure 17 bioengineering-11-00403-f017:**
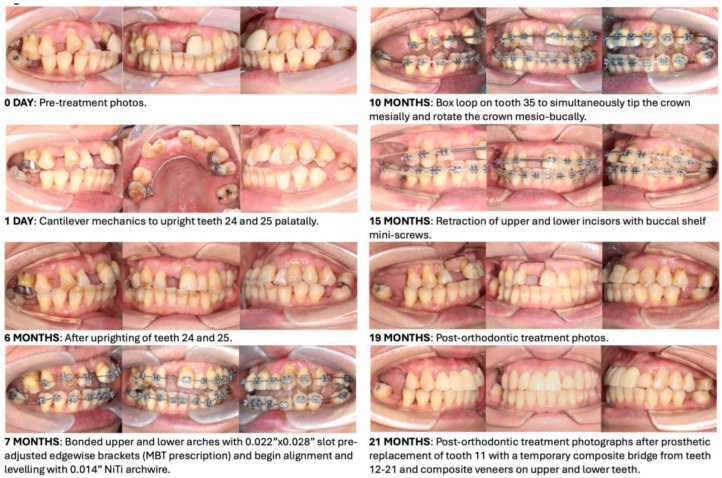
Timeline of the orthodontic treatment.

**Table 1 bioengineering-11-00403-t001:** Pre-orthodontic and near-end-of-treatment cephalometric tracing of the patient.

Variable	Norm [34]	T0	T1
SNA	82° ± 3.5	* 88.8°	* 88.5°
SNB	79° ± 3.0	* 84.1°	* 83.9°
ANB	3.0° ± 2.0	4.7°	4.6°
MMPA	26° ± 5	24.7°	24.6°
Face height ratio	55% ± 1.5	* 52.9%	* 52.8%
SN to maxillary plane	8° ± 3	* 12.1°	* 11.9°
Upper incisor to maxillary plane	118° ± 6	** 133.4°	* 129.1°
Lower incisor to mandibular plane	97° ± 7	* 105.7°	100.7°
Interincisal angle	115° ± 8	** 96.3°	* 105.6°
Wits appraisal	−4.5 mm ± 3.0	* −0.4 mm	-
Lower incisor to APo line	5.5 mm ± 2.5	** 11.4 mm	** 9.5 mm

Cook MS and Wei SHY (1988). Cephalometric standards for the Southern Chinese. Eur. J. Orthod. 10(3):264-72, [34]. Green ≤ 1 standard deviation; * Blue ≤ 2 standard deviation; ** Red > 2 standard deviation.

## Data Availability

All the datas are presented in this manuscript.

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
