# Peer review of "Biomechanical Considerations in the Orthodontic Treatment of a Patient with Stabilised Stage IV Grade C Generalised Periodontitis: A Case Report"

_bioengineering, 2024, doi:10.3390/bioengineering11040403_

Round 1

Reviewer 1 Report

Comments and Suggestions for Authors

Introduction 

Successful orthodontic treatment in patients with periodontal disease requires a multidisciplinary approach involving collaboration among periodontists, orthodontists, and sometimes prosthodontists and maxillofacial surgeons.

M&M

How was the stabilization of the periodontal condition obtained?

Periodontal records are necessary to evaluate this as a reader

Discussion :

This case report provides valuable insights into the assessment, planning, and treatment of a patient with stabilized Stage IV grade C generalized periodontitis accompanied by pathological tooth migration (PTM), highlighting the special biomechanical designs adopted. Despite meticulous planning, several challenges were encountered during treatment.

The first challenge pertained to the bulkiness of the CoCr plate covering teeth 24 and 25, hindering the patient's cleaning efficiency and predisposing the area to food entrapment. Emphasizing the importance of meticulous oral hygiene in this area became imperative. The second challenge arose from the attempted placement of mini-screws on the upper alveolar bone, resulting in dislodgement due to the low sinus floor. To address this, class I elastics were used to retract the upper arch, leveraging the lower buccal shelf mini-screws for anchorage. The third challenge involved a supernumerary tooth, which the patient refused to have extracted, affecting the planned retraction of upper incisors.

Control of orthodontic force level and direction is paramount in such cases, considering the compromised periodontal support and altered center of resistance (Cres). The use of a box loop was instrumental in delivering the desired force magnitude and direction, while also providing adequate anchorage. Bonding brackets cervically and employing sectional cantilever mechanics were strategic decisions to mitigate adverse force systems and accommodate the buccal displacement of teeth 24 and 25, ensuring the planned arch form was achieved.

Overall, successful management of periodontally compromised patients undergoing orthodontic treatment necessitates a holistic evaluation of periodontal and orthodontic factors, interdisciplinary collaboration in treatment planning, and meticulous attention to anchorage and biomechanics to ensure optimal treatment outcomes.

References:

Albandar JM, Rams TE. Global epidemiology of periodontal diseases: an overview. Periodontol 2000. 2002;29:7-10.

Wennström JL, Stokland BL, Nyman S, Thilander B. Periodontal tissue response to orthodontic movement of teeth with infrabony pockets. Am J Orthod Dentofacial Orthop. 1993;103(4):313-319.

Comments on the Quality of English Language

minor issues are detected

Reviewer 2 Report

Comments and Suggestions for Authors

According to this manuscript, I would like to express my thanks to the authors for their efforts; it needs a revision before evaluating the possibility of publication. I would like to pay attention to the following comments:

  • It is advisable to also provide the other name of the disease, which was formerly known as generalized aggressive periodontitis.
  • What about the patient's previous history in the treatment of Stage IV, grade C generalized periodontitis?
  • The author should describe in brief the difficulty of orthodontic treatment at the age of 59 years old.
  • The author should mention the difficulty in orthodontic treatment of patients with Stage IV grade C generalized periodontitis.
  • What precautions should be taken into consideration in the current case?
  • The classical treatment protocol using antibiotics should be mentioned if it was included in the treatment.
  • The author should be describing the effect of the inflammation on orthodontic tooth movement through the remodeling process.
  • The potential effect of fixed orthodontic appliances on the formation of niches and the accumulation of dental biofilm should be addressed.
  • It is recommended, if possible, to add a flowchart illustrating how orthodontic therapy was done.
  • What about the presence of any systemic diseases?
  • It is suggested to follow the guidelines described by Levin et al. as presented in the following article discussion: "Orthodontic treatment of patients with severe (stage IV) periodontitis.".

https://www.sciencedirect.com/science/article/pii/S1073874624000057
